# Polymeric Nanoparticles Based on Tyrosine-Modified, Low Molecular Weight Polyethylenimines for siRNA Delivery

**DOI:** 10.3390/pharmaceutics11110600

**Published:** 2019-11-12

**Authors:** Alexander Ewe, Sandra Noske, Michael Karimov, Achim Aigner

**Affiliations:** 1Rudolf-Boehm-Institute for Pharmacology and Toxicology, Faculty of Medicine, Clinical Pharmacology, Leipzig University, 04107 Leipzig, Germany; alexander.ewe@medizin.uni-leipzig.de (A.E.); sandra.noske@medizin.uni-leipzig.de (S.N.); michael.karimov@medizin.uni-leipzig.de (M.K.); 2Faculty of Chemistry, Technical University Kaiserslautern, 67663 Kaiserslautern, Germany

**Keywords:** poly(ethylene) imine, PEI, RNA, siRNA delivery, tyrosine-modification, tumor xenograft

## Abstract

A major hurdle for exploring RNA interference (RNAi) in a therapeutic setting is still the issue of in vivo delivery of small RNA molecules (siRNAs). The chemical modification of polyethylenimines (PEIs) offers a particularly attractive avenue towards the development of more efficient non-viral delivery systems. Here, we explore tyrosine-modified polyethylenimines with low or very low molecular weight (P2Y, P5Y, P10Y) for siRNA delivery. In comparison to their respective parent PEI, they reveal considerably increased knockdown efficacies and very low cytotoxicity upon tyrosine modification, as determined in different reporter and wildtype cell lines. The delivery of siRNAs targeting the anti-apoptotic oncogene survivin or the serine/threonine-protein kinase PLK1 (polo-like kinase 1; PLK-1) oncogene reveals strong inhibitory effects in vitro. In a therapeutic in vivo setting, profound anti-tumor effects in a prostate carcinoma xenograft mouse model are observed upon systemic application of complexes for survivin or PLK1 knockdown, in the absence of in vivo toxicity. We thus demonstrate the tyrosine-modification of (very) low molecular weight PEIs for generating efficient nanocarriers for siRNA delivery in vitro and in vivo, present data on their physicochemical and biological properties, and show their efficacy as siRNA therapeutic in vivo, in the absence of adverse effects.

## 1. Introduction

The induction of RNA interference (RNAi) by small interfering RNAs (siRNAs) [1] has great potential, allowing for the downregulation of pathophysiologically over-expressed genes on the post-transcriptional level. Since siRNAs can be specifically directed against virtually any target gene, including otherwise “undruggable” genes, they offer novel avenues towards customized therapies against many diseases, for example, the treatment of cancer (see, e.g., [2] for review). Today, siRNAs can be easily designed and chemically synthesized to target a gene of interest in a sequence-specific manner. By harnessing the cellular enzyme machinery, siRNAs are incorporated in the RNA-induced silencing complex (RISC) and, upon activation, the siRNA guide strand binds to its complementary target mRNA, leading to mRNA cleavage, degradation, and thus long-lasting gene knockdown. This is a unique characteristic of RNAi which cannot be achieved with small molecules or anti-sense oligonucleotides [3,4].

While it is thus necessary and sufficient to deliver siRNAs to the cell for inducing RNAi-mediated gene knockdown, the physicochemical properties of siRNAs, i.e., relatively high molecular weight, high polyanionic charge density, hydrophilic properties, and their instability/susceptibility to nuclease degradation, largely prevent their use in an unmodified form for therapeutic approaches [5]. Beyond chemical modifications of the siRNA, delivery systems are required [6,7]. Apart from siRNA conjugates specifically suited for liver uptake, a variety of different cationic lipids, polymeric materials, and other nanoscale systems have been developed to adsorb, electrostatically interact with or incorporate siRNA and mediate cellular uptake [8,9,10,11,12]. For in vitro application several compounds have been commercialized as transfection reagents and are able to induce a potent gene knockdown at low siRNA concentrations in cell culture [13]. However, they are generally not suited for in vivo applications and clinical translation is still challenging, and thus require the development of novel delivery systems. On the other hand, the feasibility of siRNA therapeutics has been shown when the first siRNA drug, ONPATTRO^®^ (Patisiran), was approved by the US Food and Drug Administration (FDA) and European Medicines Agency (EMA) in 2018. ONPATTRO^®^ is a lipid-based siRNA nanoparticle designed to downregulate transthyretin in the liver for treatment of the polyneuropathy of hereditary transthyretin-mediated amyloidosis (hATTR) [14,15]. Despite this success, it might be assumed that one delivery platform cannot address all diseases, especially when it comes to target tissues other than the liver [16].

Polymeric reagents are another well studied class of materials for the delivery of nucleic acids (for review, see [17,18]). A highly investigated cationic polymer is polyethylenimine (PEI), which is able to efficiently deliver nucleic acids in vitro and in vivo. PEIs are available in branched and linear topologies over a wide range of molecular weights (0.8–100 kDa) (for review, see [19,20]). While higher molecular weight PEIs show biological activity but are associated with toxicity, low molecular weight PEIs are more biocompatible but essentially lack transfection efficacy, especially in the case of smaller oligonucleotides like siRNA. Branched PEIs are more efficient for the delivery of siRNA than their linear counterparts [21,22,23]. Since the commercially available 25 kDa branched PEI shows some efficacy but is also attributed to toxic effects, the lower molecular weight PEI F25-LMW (branched; 4–10 kDa) has been introduced for siRNA delivery in vitro and in vivo [24]. Still, further improvements are warranted and various approaches have been explored to optimize transfection efficacies of low molecular weight PEIs and/or to decrease the toxicity of high molecular weight PEIs. These include the covalent modification of PEI with polyethylene glycol (PEG) [25,26,27], the cross-linking of small PEIs with hydrolysable linkers, e.g., esters, ketals, or reductive cleavable disulfide groups leading to the reversible formation of higher molecular weights ([28,29,30,31]; see [32] for review), the copolymerization/grafting with other polymers like polycaprolactone (PCL), starch or polyvinylalcohol (PVA) [33,34,35], the modification with lipophilic molecules, e.g., fatty acids, alkanes or cholesterol ([36,37], see [38] for review), or the combination of PEI-based complexes with liposomes [39,40,41,42].

The chemical modification of PEI with amino acids offers another very promising approach for nucleic acid delivery in vivo. In previous studies, the grafting of arginine, lysine, or leucine onto PEI showed increased β-galactosidase activity in tumor-bearing mice upon i.v. injection of modified PEI/pDNA complexes [43]. In another report, a library of twenty amino acids conjugated to a poly(amidoamine) (PAMAM G5) dendrimer was characterized for pDNA delivery in vitro and identified cationic and hydrophobic amino acids as most effective [44]. The triple modification of PAMAM G5 with arginine, histidine, and phenylalanine revealed synergistically enhanced pDNA reporter gene expression in vitro and in vivo as compared to their mono-functionalized derivatives [45]. Amino acid modifications have also been shown to improve the delivery of siRNA. Derivatives of a branched 25 kDa PEI with leucine, tryptophan, phenylalanine, or tyrosine were evaluated in vitro and identified the tyrosine-modified PEI as efficient for siRNA transfection [46]. The same high molecular weight 25 kDa branched PEI modified with *N*-acetyl-leucine was used for the delivery of miR34a, significantly improving bone regeneration in an orthodontic in vivo model [47]. The incorporation of histidine and tyrosine also improved in vitro siRNA transfection efficacy in the case of oligoamino amides, a new class of polymers prepared by solid phase synthesis, and decreased the EG5 mRNA expression upon i.v. injection of siEG5 nanoparticles in tumor bearing mice [48,49].

It becomes apparent from these studies that for the successful modification of cationic polymers with amino acids a balanced hydrophobicity and hydrophilicity is of great importance. This may then also allow for exploring lower molecular weight PEIs which are insufficiently active in the unmodified form, especially for the delivery of small nucleic acid molecules like siRNAs, but show significantly improved bioactivities upon introducing modifications. In a recent study, we selected a low molecular 10 kDa branched PEI for tyrosine modification. This new PEI derivative (P10Y) showed excellent in vitro siRNA transfection efficacies and increased complex stability upon serum contact. Moreover, P10Y/siRNA complexes decreased tumor growth in an aggressively growing melanoma xenograft mouse model [50,51]. Based on these results, we extended this approach towards even smaller 2 kDa and 5 kDa branched PEIs. Initial data indicated high knockdown efficacies of these small tyrosine-modified derivatives compared to their otherwise inactive parent PEIs [51]. In this study, we comprehensively explore these small tyrosine-PEIs in various cell lines, for targeting different reporter genes as well as endogenous genes. We identify P5Y as particularly active, and thus also studied this most efficient derivative in a tumor xenograft mouse model.

## 2. Materials and Methods

### 2.1. Cells and Cell Culture

Cell culture plastics and consumables were from Sarstedt (Nümbrecht, Germany). Cell culture media were from Sigma-Aldrich (Taufkirchen, Germany) and fetal calf serum was purchased from Biochrom (Berlin, Germany). The cell lines HCT116 (colorectal carcinoma), H441 (lung adenocarcinoma), PC3 (prostate carcinoma), Saos-2 (osteosarcoma), and MV3 (melanoma) were obtained from ATCC/LGC Promochem (Wesel, Germany), and G55T2 (glioblastoma) was a kind gift from Katrin Lamszus (University Clinic Hamburg-Eppendorf, Hamburg, Germany). All cell lines were cultivated in a humid atmosphere at 37 °C and 5% CO_2_. HCT116, PC3, G55T2 cells were grown in IMDM medium, H441 and MV3 cells were grown in RPMI 1640 medium, and Saos-2 cells were grown in McCoy’s 5a medium. All media were supplemented with 10% FCS and 2 mM alanyl-glutamine.

Reporter cell lines were stably transduced by lentiviral transduction. The EGFP/luciferase plasmid pCCLc-MNDU3-Luciferase-PGK-EGFP-WPRE was a gift from Fernando Fierro (University of California, Davis, CA, USA; Addgene plasmid # 89608) [52]) and the luciferase plasmid pLenti PGK V5-LUC Puro (w543-1) was a gift from Eric Campeau and Paul Kaufman (University of Massachusetts, Worchester, MA, USA; Addgene plasmid # 19360) [53]. Cell lines were cultivated in the above mentioned media and stable reporter cells were selected by either using puromycin (luciferase plasmid) or sorted by FACS (EGFP/luciferase plasmid).

### 2.2. Synthesis of Tyrosine-Modified PEIs

Branched polyethylenimines were obtained from the following suppliers: 2 kDa PEI (Sigma-Aldrich, Taufkirchen, Germany), 5 kDa PEI (a kind gift from BASF, Ludwigshafen, Germany), and 10 kDa PEI (Polysciences, Eppelheim, Germany). *N*-Boc-tyrosine-OH, *N*-hydroxysuccinimide, EDC·HCl were from Carbolution Chemicals (Saarbrücken, Germany). Dry *N*,*N*-Dimethylformamide (DMF) was from VWR (Darmstadt, Germany), Trifluoroacetic acid (TFA), ethanol, and methanol were from Carl Roth (Karlsruhe, Germany).

For the tyrosine modification, *N*-Boc (*tert*-butyloxycarbonyl) protected tyrosine (0.65 g, 2.3 mmol) and *N*-hydroxysuccinimide (0.27 g, 2.3 mmol) were dissolved in 3 mL dry DMF in a glass vial, followed by addition of EDC·HCl (0.45 g, 2.3 mmol), and stirred for 4 h under a nitrogen atmosphere at RT. In a second vial, PEI (0.2 g, 4.65 mmol in ethylenimine) was dissolved in 3 mL dry DMF, and the pre-activated tyrosine mixture was added and further stirred under nitrogen for 3 d. The reaction mixture was purified by dialysis (1 kDa MWCO regenerated cellulose, Serva, Heidelberg, Germany) against methanol for 10 h with intermediate solvent replacement to remove by-products. Next, the methanol was removed in vacuo and the viscous polymer was dissolved in 3 mL TFA, prior to stirring overnight for Boc-deprotection. Excess TFA was removed by co-evaporation with ethanol. Finally, the crude polymer was dissolved in 0.1 M HCl and excessively purified by dialysis against 0.05 M HCl for 24 h, then against water for 48 h. Lyophilization yielded the tyrosine-modified PEIs as yellowish fluffy powders. The degree of functionalization (~25–30% based on ethylenimine monomer) was confirmed by ^1^H-NMR (Mercury plus, 300 MHz, Varian Agilent Technologies, Santa Clara, CA, USA) as described previously [50,51].

### 2.3. Polyplex Preparation

The polyplexes were prepared based on polymer/siRNA mass ratios. Typically, the P5Y- and P10Y-based complexes were prepared at a mass ratio of 2.5 unless indicated otherwise. For P2Y, a mass ratio of 10 was used. For standard transfection experiments in a 24 well plate, 30 pmol (0.4 µg) siRNA per well was diluted in 12.5 µL HN buffer (150 mM NaCl, 10 mM HEPES (4-(2-hydroxyethyl)-1-piperazineethanesulfonic acid), pH 7.4). In a separate tube, the calculated amount of the polymer was diluted in 12.5 µL HN buffer and the siRNA solution was added to the polymer solution, thoroughly mixed by pipetting up and down and incubated for 30 min at RT. For serum stability studies, the prepared polyplexes were additionally incubated with different volumes of FCS.

For in vivo experiments, the polyplexes were prepared as above, with slight modifications. Per dose for injection, 10 µg siRNA was diluted in 75 µL HN buffer and 25 µg (mass ratio 2.5) of the polymer (P5Y) was diluted in 75 µL 5% (*w/v*) glucose, 10 mM HEPES pH 7.4. The complexation was performed as described above.

### 2.4. Polyplex Characterization

The hydrodynamic diameters and zeta potentials were measured by dynamic light scattering (DLS) and phase analysis light scattering (PALS) with a Brookhaven ZetaPALS system (Brookhaven Instruments, Holtsville, NY, USA). Polyplexes containing 5 µg siRNA in 250 µL were prepared as described above and diluted to 1.7 mL with millipore water. The data were analyzed using the manufacturer’s software, with applying the viscosity and refractive index of pure water at 25 °C. For the size determination, polyplexes were measured in five runs with a run duration of 1 min per experiment. Zeta potentials were analyzed in ten runs, with each run containing ten cycles using the Smoluchowski model. Additionally, polyplex sizes were analyzed by nanoparticle tracking (NTA) using a NanoSight LM10 (Malvern, Wiltshire, UK) equipped with a 640 nm sCMOS camera, software NTA 3.0, and a siRNA concentration of 1 µg/mL.

Complex stabilities were determined by a heparin displacement assay. The complexes based on the different tyrosine-modified PEIs were prepared at their optimal mass ratios as described above, prior to further incubation in FCS (50%, *v/v*) for 1 h. The complexes were then subjected to the polyanion heparin at different concentrations (units), as indicated in Appendix A. A total of 25 µL of the mixture containing 0.2 µg siRNA was analyzed by 1.5% agarose gel electrophoresis.

### 2.5. Cell Transfection

For standard transfection experiments, the cells were seeded at a density of 35,000 cells per well of a 24 well plate in 0.5 mL fully supplemented medium. The next day, the cells were transfected with polyplexes containing 30 pmol/0.4 µg siRNA per 24 well as described above in the presence of FCS and without further medium change. For siRNA targets and sequences see Appendix A.

### 2.6. Determination of Knockdown Efficacies and Complex Uptake

Luciferase activities were determined 72 h after transfection using the Beetle-Juice Kit (PJK, Kleinblittersdorf, Germany). The medium was aspirated and the cells were lysed with 300 µL Luciferase Cell Culture Lysis Reagent (Promega, Mannheim, Germany) for 30 min at RT. In a test tube, 10 µL lysate was mixed with 25 µL substrate and the luminescence was immediately measured in a luminometer (Berthold, Bad Wildbad, Germany).

The EGFP expression was determined by flow cytometry after 72 h. The cells were trypsinized, centrifuged for 3 min at 3000 rpm, and resuspended in 0.5 mL PBS, 1% BSA. The cells were measured in an Attune^®^ Acoustic Focusing Cytometer (Applied Biosystems, Foster City, CA, USA). Finally, 20,000 events in the vital gate were analyzed using the Attune^®^ software (V2.1.0).

For uptake experiments, complexes were prepared as described above (Section 2.4), but using a fluorophore-labeled siRNA-Atto488. PC3 wildtype cells were transfected, after 24 h prepared as described above with an additional PBS washing step and analyzed by flow cytometry.

To visualize the complex uptake by confocal microscopy, PC3 wildtype cells were seeded at a density of 200,000 cells onto glass coverslips in a 6 well plate. The next day, the cells were transfected with complexes of the different tyrosine-modified PEIs containing Alexa Fluor ^®^ 647-labeled siRNA (2 µg siRNA/well). After 36 h, the siRNA uptake and intracellular distribution was analyzed by confocal microscopy (TCS SP8, Leica, Wetzlar, Germany).

### 2.7. Cell Proliferation and Viability Assays

For proliferation assays, cells were seeded at a density of 2000 cells (PC3 and Saos-2) or 1000 cells (MV3) per well of a 96 well plate in 100 µL medium on the day before transfection. For endpoint cell viability assays, PC3 or G55T2 cells were seeded at a density of 7000 cells per 96 well in 100 µL medium. The number of metabolically active cells after transfection was determined by using a colorimetric assay (WST-8 Cell Counting Kit-8; Dojindo Molecular Technologies EU, Munich, Germany). Briefly, after aspirating the medium, 50 µL of a 1:10 dilution of WST-8 in serum-free medium was added to the cells and incubated at 37 °C for 1 h. The absorbance was measured at 450/620 nm in a plate reader.

Acute cell damage caused by the polyplexes was determined by measuring the extent of lactate dehydrogenase (LDH) release, using the Cytotoxicity Detection Kit (Roche, Mannheim, Germany) according to the manufacturer’s manual. Briefly, conditioned medium from transfection experiments (negative control siRNA) was collected after 24 h. For the determination of the maximum LDH release, cells were lysed with Triton X-100 at a final concentration of 2%, and conditioned medium from untreated cells served as negative control. In a 96 well plate, 50 µL sample medium was mixed with 50 µL reagent mix and incubated for 30 min in the dark. The reaction was stopped with 50 µL 1 M acetic acid and the absorption at 490/620 nm was measured using a plate reader. For blank value correction, fresh fully supplemented medium was mixed with the reagent and subtracted from all values.

### 2.8. RNA Preparation and Quantitative RT-PCR (RT-qPCR)

Total RNA was isolated using a combined TRI reagent and silica column protocol. Per 24 well, cells were lysed in 500 µL TRI reagent (TriFast, VWR, Darmstadt, Germany) and incubated for 5 min at RT. The lysate was transferred into a 1.5 mL tube and 200 µL chloroform was added, mixed well by shaking and incubated for 5 min at RT. The samples were centrifuged at 12,000 g for 15 min and the upper aqueous phase (~300 µL) was pipetted into a new 1.5 mL tube containing 300 µL 100% ethanol. After mixing, the sample was loaded onto the column (Zymo-Spin IC, Zymo Research, Freiburg, Germany), prior to centrifugation at 8000 g for 1 min. The flow-through was discarded. The column was washed twice with 450 µL 3 M sodium acetate pH 5.2, and once with 70% ethanol by centrifugation at 8000 g for 1 min; the collection tube was emptied after each step. For the drying step, the collection tube was replaced by a new one, and the columns were centrifuged at 12,000 g for 2 min. For RNA elution, 15 µL DEPC-water, pre-warmed to 65 °C, was added onto the column and incubated for 2 min, before finally eluting the RNA by centrifugation at 8000 g for 2 min. RNA concentrations and purities were measured in a NanoDrop 2000c (Thermo Fisher, Schwerte, Germany).

The total RNA was reverse transcribed with the RevertAid™ H Minus First strand cDNA synthesis Kit (Thermo Fisher, Waltham, MA, USA). In brief, 1 µL random hexamer primer was added to 1 µg RNA diluted in 10 µL DEPC-treated water, followed by an incubation step for 5 min at 65 °C and cooling to 4 °C. Then, 4 µL 5× reaction buffer, 2 µL 10 mM dNTP mix, 2.5 µL DEPC-treated ddH_2_O, and 0.5 µL RevertAid™ H Minus M-MuLV Reverse Transcriptase (200 U/µL) were added. The cDNA synthesis mixture was incubated at 25 °C for 10 min, 42 °C for 60 min, and heat denatured at 70 °C for 10 min.

For quantitative real time PCR, a StepOnePLUS Real-Time PCR System (Applied Biosystems, Foster City, CA, USA) was used, applying the ΔΔCt method [54]. The cDNA was 1:10 diluted with DEPC-water and 4 µL were mixed with 5 µL 2x PerfeCTa SYBR^®^ Green FastMix ROX (Quantabio, Beverly, MA, USA) and 1 µL 5 µM forward and reverse primer mix (see Appendix A for primer sequences). The qPCR were run with a pre-incubation at 95 °C for 2 min, followed by 45 amplification cycles (95 °C for 15 s, 55 °C for 15 s, 72 °C 15 s).

### 2.9. Hemolysis

To determine the hemolytic activity of the complexes, whole blood from healthy mice was diluted in Ringer’s solution and erythrocytes were purified by several centrifugation and washing steps at 5000 rpm for 5 min, until the supernatant became clear. After the final washing step, the cells were taken up into physiological NaCl solution. Then, 1 × 10^6^ cells in 50 µL were mixed with 50 µL complexes containing different siRNA amounts and incubated for 1 h at 37 °C. For absorption measurements, 25 µL of each sample were transferred into a 96 well plate in triplicates. Erythrocytes incubated with buffer served as negative control and erythrocytes treated with Triton X-100 to a final concentration of 1% served as positive control (= 100%). The absorbance of the samples was measured at 550 nm using a plate reader.

To evaluate erythrocyte aggregation, 50 µL of the purified cells (~1 × 10^6^ cells/mL) were mixed with 50 µL aqueous solution containing 7.5 µg P5Y/3 µg siCtrl, or 750 kDa branched PEI (1 and 3 µg) as positive control. After incubation for 2 h at 37 °C, the samples were mounted onto glass slides and examined microscopically.

### 2.10. In Vivo Tumor Therapy

Athymic nude mice (Foxn1nu, Charles River Laboratories, Sulzfeld, Germany) were kept at 23 °C in a humidified atmosphere, 12 h light/dark cycle, with standard rodent chow and water ad libitum. Animal studies were performed according to the national regulations and approved by the local authorities (Landesdirektion Sachsen, approval NO. TVV 38/16, date: 20 January 2017). Tumor xenografts were established by injecting 3 × 10^6^ PC3 cells in 150 µL PBS subcutaneously (s.c.) into both flanks of mice. When tumors reached a size of ~100 mm^3^, the mice were randomized into different treatment groups (6–9 tumors/group). The polyplexes were prepared as described above and amounts corresponding to 10 µg siRNA were i.p. injected every 2–3 days over 14 days.

For in vivo luciferase knockdown experiments, 3 × 10^6^ HCT116-Luc cells were s.c. injected for the induction of tumor xenografts described above. Mice were treated five times every 2–3 days with P5Y/siRNA complexes (siLuc2 = siCtrl and siLuc3 = Luciferase specific siRNA). Tumors were excised and lysed with Luciferase Cell Culture Lysis Reagent (app. 100 mg tumor tissue/1 mL buffer), prior to homogenization using an ULTRA-TURRAX^®^. Homogenates were repeatedly centrifuged at 13,000 rpm for 5 min and the supernatants were transferred into fresh tubes until the lysate became clear. Luminescence was determined as described above. Relative light units (RLU) were normalized for total protein concentration of the lysates using the BCA assay (Pierce Thermo Fisher, Schwerte, Germany) according to the manufacturer’s protocol.

For the determination of blood serum markers, healthy nude mice were i.p. injected with P5Y/siLuc3 complexes containing 10 µg siRNA every 2–3 days, with a total of four times over 8 days. Three hours after the last injection, the blood was collected. Untreated mice served as negative control. The serum was diluted 1:20 with water and serum levels of various analytes were determined using an AU480 (Beckman Coulter, Krefeld, Germany).

For the analysis of the immunostimulatory cytokines TNF-α and INF-γ, P5Y/siLuc3 complexes (10 µg siRNA) were i.v. injected twice within 24 h into immunocompetent C57BL/6 mice. Four hours after the last injection, the blood was collected. Mice treated with lipopolysaccharides (LPS) 50 µg in 150 µL (single injection) served as positive control and untreated mice served as negative control. The serum levels of TNF-α and INF-γ were measured using ELISA kits (PreproTech, Hamburg, Germany) following the manufacturer’s instructions.

### 2.11. Statistics

Statistical analyses were performed by Student’s *t*-test or One-way ANOVA, and significance levels are * = *p* < 0.05, ** = *p* < 0.01, *** = *p* < 0.001, and # = not significant, with at least *n* = 3.

## 3. Results

### 3.1. Identification of Optimal Complexes for In Vitro Transfection

Based on the superior siRNA complexation and transfection efficacy of 10 kDa PEI upon its tyrosine-modification, we analyzed the performance of even smaller PEIs, which had been shown previously to be inactive [51], upon tyrosine engraftment. We hypothesized that even in the case of 5 kDa or 2 kDa PEIs, the tyrosine modification may result in polymers capable of efficient siRNA complexation and delivery, thus leading to target gene knockdown and RNAi-mediated tumor cell-inhibitory effects in various cellular assays. The tyrosine was covalently coupled onto branched PEIs by using standard NHS/EDC coupling chemistry as outlined in Figure 1A. The degree of tyrosine modification was ~30% compared to ethylenimine monomers, determined by ^1^H-NMR analysis as described ([50,51]; see Appendix A). Transfection experiments with complexes based on the tyrosine-modified 5 kDa PEI (P5Y) revealed >90% knockdown of serine/threonine-protein kinase PLK1 (polo-like kinase 1; PLK-1) when transfecting a specific siPLK1 as compared to a negative control siRNA (siCtrl; Figure 1B, left). Complexes based on P10Y or P5Y were even more efficient than their higher molecular weight 25 kDa counterpart (P25Y; data not shown), while P2Y/siRNA complexes comprising the very low molecular weight 2 kDa PEI were less efficient. Similar results were obtained after transfection of an siRNA targeting the anti-apoptotic protein survivin, with P5Y/siRNA and P10Y/siRNA complexes again resulting in a 80–90% knockdown (Figure 1B, right) and thus performing better than their P25Y/siRNA counterparts (not shown). P2Y-based nanoparticles showed somewhat reduced activity, even when using higher N/P ratios. The knockdown of the oncogenes/proto-oncogenes PLK-1 and survivin also exerted profound tumor cell-inhibitory effects in anchorage-dependent proliferation assays (Figure 1C). When using P5Y or P10Y, the transfection of 6 pmol/well siRNA was sufficient to largely abolish cell proliferation. In contrast, in the case of P2Y/siRNA complexes, even double amounts (12 pmol) were incapable of inhibiting cell growth. Only the transfection of a very potent ubiquitin siRNA, which essentially abolished tumor cell viabilities in the P5Y and P10Y transfection groups, was strong enough to partially inhibit cell growth when complexed with P2Y. For the very efficient P10Y and P5Y, the siRNA concentration could be further reduced to 3 pmol siRNA with still showing strong inhibitory effects (Appendix A). In addition, this reduction revealed differences in the potencies of the siRNAs against different target genes.

To further analyze possible reasons for these observed differences in biological activity, we determined intracellular levels of fluorophore-labeled siRNAs upon transfection of PC3 cells complexed with the different tyrosine-modified PEIs. The comparison of P5Y/atto488-siRNA complexes with their P10Y counterparts used for transfection in the presence of fetal calf serum (FCS) revealed a ~30% decrease in siRNA fluorescence when switching from P5Y to P10Y, while siRNA delivery in P2Y complexes was poor (Appendix A). This was also confirmed in confocal microscopy, demonstrating atto488-siRNA signals to be largely absent upon transfection with P2Y (Appendix A, upper panel). In contrast, profound fluorescence was observed in the case of P5Y and P10Y, with somewhat stronger signals in the case of P5Y/siRNA complexes (Appendix A, center and lower panels). Beyond complex uptake, these differences may be based on variations in complex stabilities. Interestingly, a heparin displacement assay revealed even enhanced P5Y/siRNA complex stability in the presence vs. in the absence of serum, as indicated by the shift towards higher heparin concentrations required for siRNA release from the complex upon its pre-incubation in FCS (Appendix A, upper panels). Under the same conditions, siRNA release from P2Y/siRNA complexes was already observed at lower heparin concentrations, indicating poor complex stability. In stark contrast, siRNA bands from heparin displacement of P10Y/siRNA complexes were weaker, independent of the heparin concentration, and thus indicating incomplete siRNA release even under very stringent conditions. Taken together, this suggests differences in siRNA delivery, complex stability, and siRNA release as underlying reasons for molecular weight-dependent differences between the tyrosine-modified PEIs. Very high efficacies of tyrosine-modified branched PEIs were observed up to as low as 5 kDa (P5Y), while only an even further reduction of the molecular weight (P2Y) eventually led to reduced transfection efficacies. This prompted us to focus in particular on P5Y as the tyrosine-modified PEI with the lowest molecular weight and highest activity, which was even slightly above P10Y.

The very profound siRNA transfection efficacies of P5Y were also confirmed in other cell lines like Saos-2 osteosarcoma cells (Figure 1D). A 1.5-fold increase of the polymer/siRNA mass ratio did not lead to enhanced knockdown, indicating that the very low polymer/siRNA mass ratio of 2.5 was already sufficient for maximum complex activity (Figure 1D). In contrast, knockdown efficacies were found to be dependent on the selection of the siRNA and the target gene, as indicated by the particularly profound >90% knockdown in the case of an siRNA targeting glyceraldehyde-3-phosphate dehydrogenase (GAPDH). As before in PC3 cells, an almost complete abolishment of cell proliferation was observed upon transfecting Saos-2 cells with P5Y/siPLK-1 or P5Y/siSurv complexes (Figure 1E). The microscopic evaluation of the wells also revealed a complete (siSurv) or almost complete (siPLK-1) loss of viable cells after 72 h (Figure 1E, right panel). In all experiments, P5Y-complexed negative control siRNA (siCtrl) was transfected in parallel, with the comparison to untransfected cells indicating non-specific transfection effects to be largely absent. As seen before in PC3 cells, an increase in the polymer/siRNA ratio by 1.5-fold did not further enhance biological efficacies. However, the P5Y/siCtrl curve indicated some non-specific inhibitory effects at this ratio due to excess polymer (Appendix A). Specific growth inhibition upon P5Y/siRNA-mediated survivin knockdown was also observed in MV3 melanoma cells (Appendix A), and GAPDH target gene reduction similar to the results shown above were also found in other cell lines (H441 lung adenocarcinoma cells, PC3 prostate carcinoma cells, and G55T2 glioblastoma cells; Appendix A).

### 3.2. Characterization of P5Y/siRNA Complex Properties

For physicochemical characterization, complexes with different polymer/siRNA mass ratios were subjected to Zetasizer and Nanosight measurements. The hydrodynamic diameters determined by dynamic light scattering (DLS) were around 200–400 nm, with a slight trend towards smaller complexes with lower P5Y/siRNA mass ratios (Figure 2A).

The diameter for the optimal mass ratio of 2.5 was additionally characterized by nanoparticle tracking (NTA) of which the main peak was at 200 nm and a second small peak at 400 nm (Figure 2B). The measurement of the zeta potentials revealed a slightly negative value for the lowest mass ratio of 1.25 indicating non complexed siRNA, and strongly increased to +20 mV for the mass ratio 2.5. The zeta potential further increased to ~30 mV at mass ratio 5 and reached a plateau of ~25 mV for mass ratios above 10. In many nanoparticle formulations, serum stability has been identified as one major issue limiting biological activity. Our transfection experiments, however, were exclusively performed in serum-containing media (10% FCS). Moreover, the presence of serum even proved beneficial: While PEI-based complexes tend to readily aggregate in aqueous solutions [42], our P5Y/siRNA complexes could be stored over weeks in the presence of 10% or even 50% FCS without losing activity, while 5% FCS was insufficient (Figure 2C).

### 3.3. High Biocompatibility/Absence of Toxicity of P5Y/siRNA Complexes

High biological activity may well be associated with increased cytotoxicity or other adverse effects, which were therefore assessed next. Lactate dehydrogenase (LDH) release assays, however, revealed no increase of LDH levels over background (untransfected cells). This was true for both, P5Y/siRNA and P10Y/siRNA complexes and was found in two reporter cell lines (PC3-Luc-EGFP and G55T2-Luc-EGFP cells; Figure 3A,D).

Again, the transfection led to profound target gene knockdown, indicated by 80–90% decreased EGFP activities upon siEGFP transfection as determined in flow cytometry (Figure 3B,E). Concomitantly, cell viabilities remained unaffected by P5Y/siRNA or P10Y/siRNA transfection, even after 48 h and independent of siRNA amounts, which could be increased from 4 pmol to 12 pmol without appreciable decrease in metabolic activity (Figure 3C,F). In line with the absence of adverse effects on cell membrane integrity (LDH release assay), a hemoglobin release assay in erythrocytes revealed no adverse effects. Within an almost 10-fold range of different complex amounts, no hemoglobin release over background was observed (Figure 3G). Similarly, the P5Y/siRNA complexes were tested for their erythrocyte aggregation potential. Identical complex amounts were incubated with red blood cells and analyzed under the microscope (Figure 3H). While the positive control a 750 kDa PEI led to strong aggregates at 1 µg, the P5Y complexes did not damage the red blood cells up to 3 µg complexed siRNA.

### 3.4. Therapeutic In Vivo Application of P5Y/siRNA Complexes Leads to Profound Anti-Tumor Effects

Pivotal in the development of nanoparticles for siRNA delivery is their applicability in vivo. In previous biodistribution assays, we showed that PEI/siRNA or P10Y/siRNA complexes become systemically available upon i.p. injection, and thus are able to reach organs/tissues remote from the injection site including tumors [50,55]. These studies were now extended towards P5Y/siRNA complexes. In a first experiment, P5Y/siRNA complexes were tested for reporter gene knockdown in mice bearing s.c. HCT116-Luc tumor xenografts. Prior to the in vivo application, the P5Y complexes were first evaluated in vitro for their potential to transfect this cell line. The luciferase expression was decreased to 60% at the lowest siRNA concentration of 15 pmol and further to 25% of control levels at 30 pmol siRNA (Figure 4A).

Next, mice were injected with 5 × 10^6^ tumor cells and, upon establishment of tumor xenografts, treated by repeated systemic application of 10 µg P5Y-complexed siRNA (3× every second day, intraperitoneal injection). Upon termination of the experiment at day 6 after treatment start, tumors were harvested and luciferase activities were analyzed from tumor lysates. Some variations were found between individual samples, especially in the negative control group. Still, the comparison of luciferase activities between the specific (P5Y/siLuc3) and the non-specific negative control (P5Y/siCtrl) group revealed a ~25% reduction of luciferase expression already after three injections (Figure 4B), and thus confirm that the treatment of mice with P5Y/siRNA complexes leads indeed to the knockdown of a given target gene in the tumors.

Switching to a more relevant therapy model, PC3 cells were s.c. injected into mice. Upon establishment of tumor xenografts of ~100 mm^3^ in size and with clear growth kinetics, mice were randomized into different groups. Specific treatment relied on the P5Y-complexed siRNAs targeting the oncogenic proteins PLK-1 or survivin. For P10Y complexation, only PLK1 was selected as specific siRNA, while untreated mice or mice treated with siCtrl-containing complexes served as negative controls. The comparison of the latter groups revealed the absence of non-specific effects of repeated treatments (3×/week), as indicated by identical tumor growth curves (Figure 5A,B). In contrast, the treatment with the P5Y/siPLK-1 or P5Y/siSurv (Figure 5A) or P10Y/siSurv complexes (Figure 5B) led to profound antitumor effects.

During the experiment and upon its termination, mice were also screened and tested for possible adverse effects due to the repeated i.p. administration of P5Y/siRNA complexes. Mice showed no alterations in behavior, no obvious weight loss, and no other signs of adverse effects. In addition, possible toxic effects were determined by measuring important serum parameters. Healthy mice were repeatedly i.p. injected with P5Y complexes comprising 10 µg siCtrl four times over eight days. The serum analyses revealed no increase in liver enzymes (ASAT, ALAT), blood glucose (Gluc), urea, albumin, or LDH (Figure 6A). Other side effects may include the unwanted stimulation of the innate immune system. To test for this, immunocompetent mice were treated twice within 24 h with P5Y/siCtrl complexes. In this experiment, i.v. injection was preferred over i.p. administration to achieve instantaneous 100% bioavailability, and serum levels were analyzed at 4 h after the last injection, as described previously [50]. As shown in Figure 6B, no increase in TNFα or IFNγ levels were observed. We thus conclude that P5Y/siRNA complexes are efficient and safe in vitro and in vivo.

## 4. Discussion

PEIs and other cationic polymers are promising with regard to nucleic acid delivery, but they are often associated with toxicity. This is particularly true for higher molecular weight PEIs, while their lower molecular weight counterparts suffer from poor efficacy [21,22,23]. The toxicity of high molecular weight PEIs occurs at later stages. An early toxic effect is largely due to a disturbance of the cell membrane after contact with the complexes, leading to increased LDH release and induction of necrosis [56]. The late-stage cytotoxicity was reported to be caused by a damage of the mitochondrial membrane and subsequent activation of caspases and cytochrome C release [57,58,59]. While the latter findings have been frequently reported, the genotoxic potential of PEI is controversially discussed. Both, the presence and absence of DNA-damaging potential has been reported [60,61,62,63] and, possibly explaining these discrepancies, may be dependent on the tested concentrations and selected cell lines.

The fact that PEIs offer relatively easy access to chemical modification is a particularly attractive approach for further improvement of physicochemical and biological properties of the polymer and its complexes. Tyrosine may be considered as among the best candidate amino acids for chemical modification, due to its ability to balance between hydrophilicity (primary amines necessary for nucleic acid complexation) and hydrophobicity (favoring cellular internalization and endosomal escape [44,46]). Many chemical modifications of low molecular weight PEIs in order to increase their bioactivity rely on the cross-linking to create larger PEI derivatives, the coupling of lipidic groups to facilitate micellar structures for improved nucleic acid binding, or modifications with cationic groups to increase the electrostatic interactions (see Introduction). The aromatic amino acid tyrosine plays a key role in protein/DNA interactions. Several studies have reported that tyrosine is essential for the binding to nucleic acids by hydrogen bonding and by π–π-stacking [64,65], which can provide an explanation for the substantial increase in very low molecular weight PEI bioactivity upon tyrosine modification. Generally, phenol-containing molecules are an interesting motif to positively affect a given nucleic acid delivery system. For example, the pre-incubation of siRNA with EGCG, a polyphenol from green tea, and subsequent complexation with low molecular weight polyamines ≤5 kDa (PEI, PLL, PAMAM) strongly increased the knockdown efficacy [22].

Serum stability is an important factor for nucleic acid delivery systems, and we demonstrate high serum compatibility of our complexes based on tyrosine-modified PEIs (see Results and [50]). This may be attributed to the hydroxyl group of tyrosine. In contrast to phenylalanine, tyrosine shows negligible interactions with bovine serum albumin (BSA), as shown previously by the incubation of thymine-dityrosine or thymine-diphenylalanine with BSA solutions [66]. Still, P5Y/siRNA complex stability was even slightly enhanced upon incubation in serum (see Appendix A). The stabilizing effect of hydroxyl groups was demonstrated in another study, by the modification of PEI with a hydroxyl-containing cyclic carbonate (EHDO). This modification improved serum-tolerance and reduced cytotoxicity as compared to the parent 25 kDa bPEI [67]. Similar results were reported for the Tris-modification of 25 kDa bPEI for pDNA and siRNA delivery [68].

Indeed, we could demonstrate in this study that tyrosine-modification allows for using very low, otherwise inactive PEIs. High complexation efficacies even at very low mass ratios are particularly relevant in the case of small oligonucleotides like siRNAs, considering the lack of binding cooperativity and their high rigidity [69]. Our results are in line with other studies demonstrating low toxicity and high biocompatibility of other (non-PEI) tyrosine-based polymers [70], as well as the safety of tyrosine in humans [71]. The beneficial effects of tyrosine modification thus allowed for using very small amounts of (derivatized) non-toxic low molecular weight PEIs whose bioactivity and biocompatibility critically relies on the grafting with tyrosine. Optimal complex stabilities, which need to be sufficiently high for efficient siRNA complexation/protection and sufficiently low for efficient intracellular siRNA release from the complex, are additional requirements. The potential relevance of the approach of using tyrosine-modified PEIs for translation into the clinic is emphasized by the fact that these nanoparticles were found to be efficacious also in vivo.

## Figures and Tables

**Figure 1 pharmaceutics-11-00600-f001:**
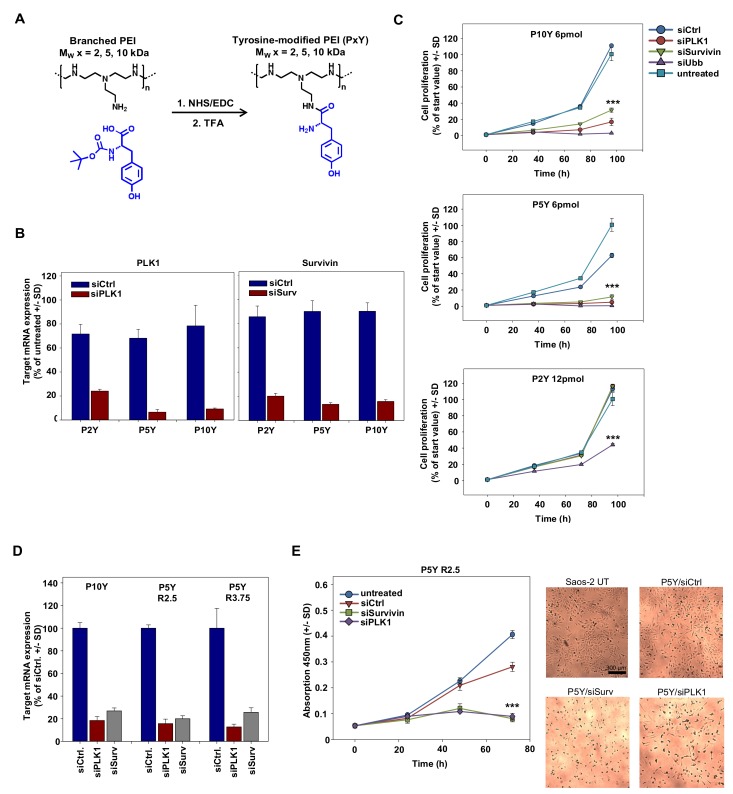
(**A**) Reaction scheme of tyrosine-coupling onto branched polyethylenimines (PEIs). Abbreviations: TFA, Trifluoroacetic acid; NHS, *N*-Hydroxysuccinimid; EDC, 1-Ethyl-3-(3-dimethylaminopropyl) carbodiimid (**B**) Knockdown efficacies of various tyrosine-modified PEI/siRNA complexes in PC3 cells targeting the oncogenes polo-like kinase 1 (PLK1) (left) or survivin (right), quantitated by RT-qPCR. (**C**) Determination of anchorage-dependent proliferation of PC3 cells upon transfection with the tyrosine-modified PEI/siRNA complexes using different anti-proliferative siRNAs at the amounts indicated in the figures. SiRNA-mediated cell inhibition is compared to negative-control transfected or untreated cells. (**D**) Knockdown upon transfecting Saos-2 cells with siRNAs against PLK1 or survivin, as determined by RT-qPCR. Increasing mass ratios from 2.5 to 3.75 does not further improve the knockdown efficacy. (**E**) Determination of anchorage-dependent proliferation of Saos-2 cells upon transfection with P5Y/siRNA complexes targeting PLK1 or survivin. The statistical significance indicates the difference to negative-control transfected cells. Left: quantitation based on WST-8 measurements; right: original pictures.

**Figure 2 pharmaceutics-11-00600-f002:**
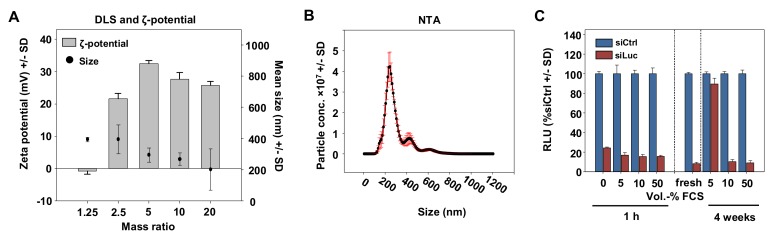
(**A**) Determination of the hydrodynamic diameters and zeta potentials of P5Y/siRNA complexes at different mass ratios as indicated in the figure. (**B**) Size measurement of P5Y/siRNA complexes by nanoparticle tracking (NTA) at the optimal mass ratio of 2.5. (**C**) Luciferase knockdown efficacies in H441-Luc cells for P5Y/siRNA complexes pre-incubated with increasing amounts of FCS (*v/v*) prior to transfection for 1 h (left) or upon storage for 4 weeks at 4 °C.

**Figure 3 pharmaceutics-11-00600-f003:**
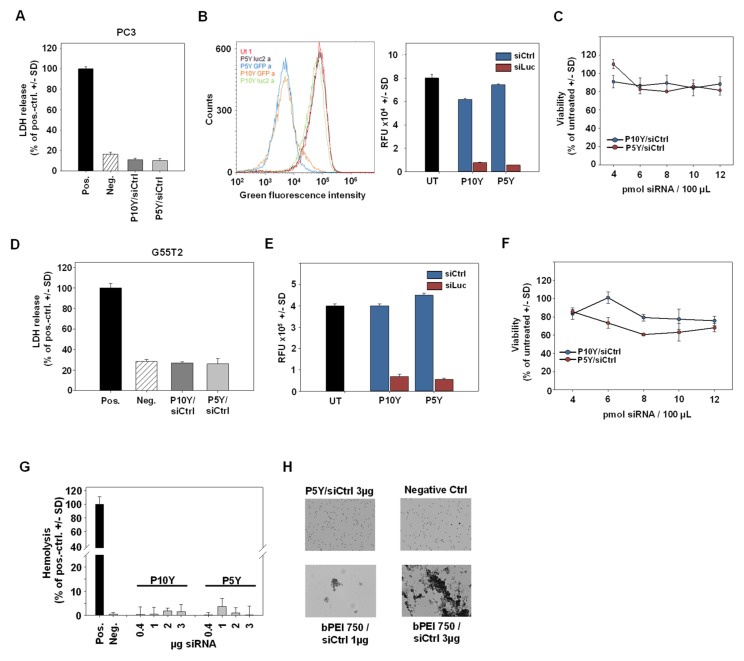
(**A**,**D**) Absence of early cytotoxic effects, as determined in lactate dehydrogenase (LDH) release assays for P10Y/siCtrl and P5Y/siCtrl complexes in PC3 (**A**) and G55T2 (**D**) cells. LDH release was quantitated 24 h after transfection of 30 pmol siCtrl/polymer per 24 well. (**B**,**E**) Knockdown of the Enhanced Green Fluorescent Protein (EGFP) reporter gene in stably expressing PC3-EGFP/Luc (**B**) and G55T2-EGFP/Luc (**E**) cells. Cells were transfected with P10Y and P5Y siRNA complexes and analyzed after 72 h by flow cytometry. (**C**,**F**) Cell viabilities of PC3 cells (**C**) and G55T2 cells (**F**) 48 h after transfection of P10Y/siCtrl and P5Y/siCtrl complexes at mass ratio 2.5 with increasing siRNA-concentrations. (**G**) Hemolysis assay for P5Y/siRNA complexes, demonstrating the absence of erythrocyte damage over a wider range of different concentrations. (**H**) Erythrocyte aggregation assay, further confirming the biocompatibility of P5Y/siRNA complexes as shown for 3 µg complexed siRNA (upper left; all lens magnifications: 4×). A total of 750 kDa bPEI complexes served as positive control, which led to a dose-dependent, profound aggregation of the red blood cells.

**Figure 4 pharmaceutics-11-00600-f004:**
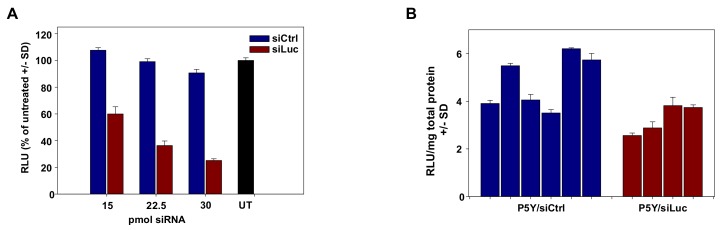
(**A**) Evaluation of luciferase knockdown efficacies of P5Y/siRNA complexes for in vivo use, in stably expressing HCT116-Luc cells over a range of different siRNA concentrations as indicated in the figure. (**B**) Luciferase activity of lysed HCT116-Luc tumors after in vivo treatment with P5Y/siRNA complexes. Bars represent the luciferase activities normalized for total protein concentrations from individual tumors of siCtrl-treated (blue) and siLuc-treated (red) mice.

**Figure 5 pharmaceutics-11-00600-f005:**
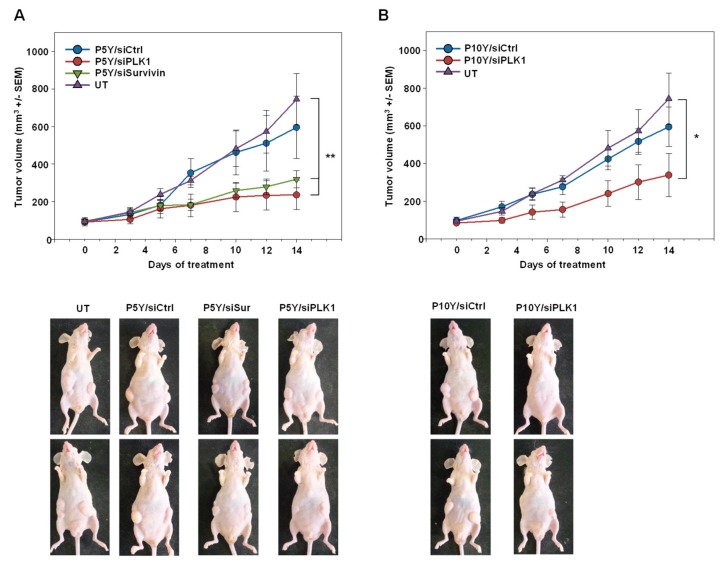
Therapy studies in PC3 tumor bearing mice using P5Y/siRNA (**A**) or P10Y/siRNA (**B**) complexes targeting either the oncogenes PLK1 or survivin. Mice were intraperitoneally treated with the complexes equivalent to 10 µg siRNA every 2–3 days. Tumor growth curves (upper panels) demonstrate the growth inhibiting effects of the specifically treated groups as compared to a negative control treated or untreated group. Lower panels: representative examples of mice.

**Figure 6 pharmaceutics-11-00600-f006:**
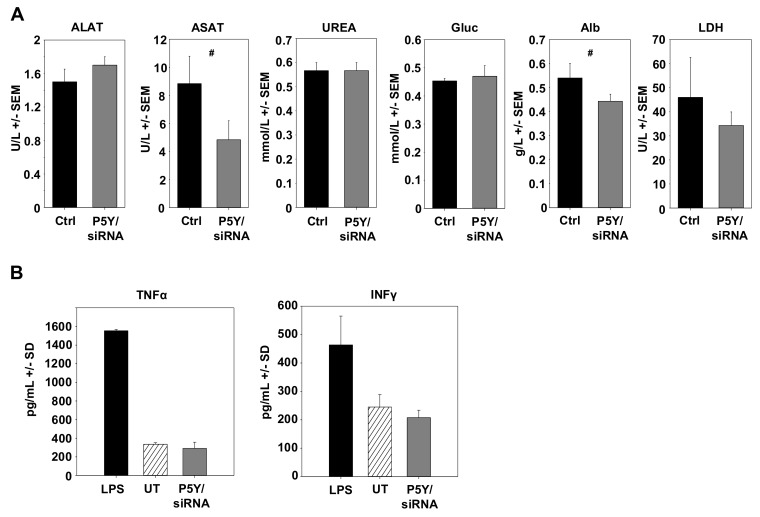
(**A**) Determination of blood serum markers for hepatotoxicity, cardiac or skeletal muscle damage, and kidney dysfunction. Healthy mice were repeatedly i.p. injected with P5Y/siCtrl complexes and blood serum levels were compared to untreated mice. ALAT: Alanine aminotransferase; ASAT: Aspartate aminotransferase; Gluc: Glucose; Alb: Albumin; LDH: Lactate deghydrogenase. (**B**) Absence of immunostimulation of P5Y/siCtrl complexes. Complexes were i.v. injected into immunocompetent mice twice within 24 h and blood was collected 4 h after the last injection. Treatment with lipopolysaccharides (LPS) served as positive control. #, not significant.

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
