# Peer review of "Polymeric Nanoparticles Based on Tyrosine-Modified, Low Molecular Weight Polyethylenimines for siRNA Delivery"

_pharmaceutics, 2019, doi:10.3390/pharmaceutics11110600_

Round 1
Reviewer 1 Report
The article titled “Polymeric nanoparticles based on tyrosine-modified, low molecular weight polyethylenimines for siRNA delivery in vitro and in vivo” describes gene silencing efficacy of PEI based polymeric formulation. Authors should address the following critical points:
-The molecular interpretation and explanation based on hypotheses are two key factors that are missing in the results & discussions part. Also, in section 2.1 authors have given a broad description of the results like a lab note-book format. The discussion has not addressed any figure in proper places. So, the readability is very poor and difficult to understand what the novelty of the present work is.
-It is advised to remove the phrase from the title; “in vitro and in vivo” as it is not properly placed.
-Characterization of complexes in section 2.2 should come first (before 2.1).
-P5L190: The statement “while polymeric complexes tend to readily aggregate in aqueous solutions” needs reference.
-Language and grammar corrections needed at some places.
Altogether, the article significantly lacks organization of the key experiments and discussions. Thus, it is suggested for a resubmission after major modification addressing the above mentioned points.
Reviewer 2 Report
This work attempted to reduce the toxicity of polyethylenimines (PEIs) for siRNA delivery, by reducing the length of PEI. Tyrosine was introduced to PEI for stabilizing the polyplex. Previously, the authors reported this concept using 10kDa PEI. Currently, they tried to reduce the PEI length and found that 5kDa is minimal length that allowed efficient siRNA delivery. Although they performed many experiments including in vivo study, overall benefit of this study seems quite low. They failed to show advantage of 5kDa PEI over 10kDa PEI. No mechanistic study was performed to explain the difference between 2kDa and 5kDa. They should address at least one of these issues. This work seems premature for publication. Please find specific comments below.
1)They failed to show advantage of 5kDa PEI over 10kDa PEI. No mechanistic study was performed to explain the difference between 2kDa and 5kDa. They should address at least one of these issues.
2) In figure 1c, PEI5k/siCtrl reduced proliferation rate compared to untreated, while PEI10k/siCtrl did not. In Figure 3F, viability after PEI5k/siCtrl delivery was lower than that after PEI10k/siCtrl delivery. These results showed, with reproducibility, that PEI5k/siCtrl is more toxic than PEI10k/siCtrl. Why?
3) l. 115, The degree of tyrosine modification was ~30 % compared to ethylenimine monomers, determined by 1H NMR analysis as described. Please provide table showing exact Tyr introduction ratio in each formulation.
4) In Figure 3G, description of horizontal axis is insufficient. Two sets of "0.4 - 3" are there. What does each of them show?
5) How did i.p. injected polyplex migrated to the tumor? How is the biodistribution profile? At least, discussion based on previous studies is needed.
Round 2
Reviewer 1 Report
The article can now be accepted for publication.
Reviewer 2 Report
Authors well addressed my comments by providing additional data. Now I recommend to accept this manuscript.